# Characterisation of Cytotoxicity-Related Receptors on γδ T Cells in Chronic Lymphocytic Leukaemia

**DOI:** 10.3390/cells14060451

**Published:** 2025-03-18

**Authors:** Michał Zarobkiewicz, Natalia Lehman, Izabela Morawska-Michalska, Adam Michalski, Wioleta Kowalska, Agata Szymańska, Waldemar Tomczak, Agnieszka Bojarska-Junak

**Affiliations:** 1Department of Clinical Immunology, Medical University of Lublin, 20-093 Lublin, Poland; lehmannatalia8@gmail.com (N.L.); izabela.morawska-michalska@umlub.pl (I.M.-M.); michalski.g.adam@gmail.com (A.M.); wioleta.kowalska@umlub.pl (W.K.); agata.szymanska@umlub.pl (A.S.); agnieszka.bojarska-junak@umlub.pl (A.B.-J.); 2Department of Haematooncology and Bone Marrow Transplantation, Medical University of Lublin, 20-080 Lublin, Poland; waldemar.tomczak@umlub.pl

**Keywords:** chronic lymphocytic leukaemia (CLL), γδ T cells, CD16, CD56, CD57, CD69, LAG-3

## Abstract

Chronic lymphocytic leukaemia (CLL) is a haematological malignancy primarily affecting older adults, characterised by the proliferation of functionally impaired B lymphocytes with abnormal expression of CD5, a typical T cell marker. The current study investigates the expression of cytotoxicity-related receptors (CD16, CD56, CD57, CD69) and a checkpoint (LAG-3) on γδ T cells in CLL patients. Sixty-nine treatment-naive CLL patients and fourteen healthy controls were recruited. Flow cytometry analysis revealed that the CLL patients had higher expressions of CD56 and LAG-3 and lower CD16 on their γδ T cells compared to the healthy controls. Subgroup analysis showed that ZAP-70-negative patients exhibited increased CD69, while CD38-negative patients showed higher CD16 expression. Additionally, CD16 expression was inversely correlated with serum LDH levels, a marker of disease progression. Bioinformatic analysis of the LAG-3 ligand mRNA in a CLL dataset indicated higher expression of *HLA-DQA2* and *HLA-DRB5* in patients with unmutated *IGVH*. Our findings highlight the altered expression of key cytotoxicity markers on γδ T cells in CLL, suggesting their potential role in disease progression and as a therapeutic target. In particular, the use of anti-LAG-3 antibodies seems promising.

## 1. Introduction

Chronic lymphocytic leukaemia (CLL) is a haematological malignancy characterised by the proliferation of morphologically mature, but functionally impaired, neoplastic lymphocytes. It primarily affects older individuals, with a median age at diagnosis of around 70 years, and over 80% of cases occur in people over 60 years [1,2]. γδT cells comprise an essential but minor subset of total human T cells. They are important for a rapid response to both infections and neoplastic processes. Human γδ T cells are classically divided based on their variable fragment of the TCRδ chain into Vδ1, Vδ2, and Vδ3 [3]. The composition of circulating γδ T cells changes with age, in infants and children, Vδ1 dominates, while in adults, Vδ2 usually comprises the majority [4]. Moreover, the overall fitness and number of γδ T cells decline with age [4,5]. Human γδ T cells recognise a set of mostly conserved antigens, for e.g., phosphoantigens (the major activator of Vδ2) and sulfatides or MIC-A/B (mostly for Vδ1 cells) [6,7].

Contrary to conventional T cells, γδ T ones recognise their targets mostly in an MHC-unrestricted manner; instead of their TCR complex, they utilise a wide range of surface receptors [8]. CD56 expression on γδ T cells is usually linked to their superior cytotoxic potential [9]. On the other hand, CD56^+^ γδT cells accumulate during healthy pregnancy, where their response seems to be tightly limited by co-inhibitory molecules like PD-1 [10]. Similarly, CD16, a low-affinity receptor for the Fc fragment of IgG, is vital for antibody-dependent cellular cytotoxicity (ADCC), a process that has therapeutic importance for CLL [11]. CD57 on T cells is usually considered a marker of terminally differentiated effector memory cells with high cytotoxic, but poor proliferative potential [12]. LAG-3 is one of the novel checkpoint molecules expressed by T cells, its expression on γδ T lymphocytes is linked with their exhaustion and dysfunction in cancer [13]. Moreover, γδ T cells expressing LAG-3 have decreased cytotoxic potential, with reduced perforin and granzyme release [13].

The expression of cytotoxicity-related receptors, like CD16, CD56, CD57, or LAG-3, on γδ T cells in CLL has not been analysed. The current paper aims to fill that void and assess the clinical importance of CD16, CD56, CD57, or LAG-3 expression on γδ T cells in CLL.

## 2. Materials and Methods

### 2.1. Participants

A total of 69 CLL patients were recruited at the Department of Haematooncology and Bone Marrow Transplantation, part of the Medical University of Lublin, between March 2016 and November 2021. The patients were treatment naive, and staging was performed according to Rai’s criteria [14]. Patients with other cancers, autoimmune diseases, or a history of immunodeficiency were excluded. The control group comprised 14 healthy individuals, matched in terms of their age and sex. Peripheral blood mononuclear cells (PBMCs) were isolated, using gradient centrifugation with Gradisol L (#9003.1, Aqua-Med, Łódź, Poland). The study protocol was approved by the Ethics Committee at the Medical University of Lublin (KE-0254/88/2016, KE-0254/176/2020), and written informed consent was obtained from all the participants. The basic sociodemographic information about the participants is presented in Appendix A.

### 2.2. Flow Cytometry

The PBMCs were stained with anti-human monoclonal antibodies: FITC anti-TCRγδ (BioLegend, San Diego, CA, USA, #331208), PE anti-CD16 (Becton Dickinson [BD], Franklin Lakes, NJ, USA, #561313), PE anti-CD56 (BD, #555516), APC anti-CD3 (BioLegend, #300439), PE-Cy7 anti-CD57 (BioLegend, #359624), APC-Cy7 anti-CD69 (BD, #557756), and APC-R700 anti-LAG-3 (BD, #565774). The cells were incubated for 30 min in the dark at 4 °C and were then washed with PBS. The samples were acquired using a Cytoflex LX analyser (Beckman Coulter, Brea, CA, USA). Finally, the FCS files were analysed with the FlowJo software, v10 version (FlowJo LLC, Ashland, OR, USA). The gating strategy is presented in Figure 1. The detailed configuration of the flow cytometer is presented in Appendix A.

### 2.3. Bioinformatics

The expression of the LAG-3 ligand mRNA (*LGALS3, FGL1, HLA-DMA, HLA-DOA, HLA-DOB, HLA-DQA1, HLA-DQA2, HLA-DRA, HLA-DRB1, HLA-DRB2, HLA-DRB3, HLA-DRB4, HLA-DRB5, HLA-DPA1, HLA-DPB1*) was analysed as previously described in regard to a publicly available dataset [15], Chronic Lymphocytic Leukemia (Broad, Nature 2015) [16]. The expression level in transcripts per million (TPM) was extracted from a subset of 157 CLL patients using the cBioPortal [17,18].

### 2.4. Statistics

The data were analysed using GraphPad Prism 9 (GraphPad Software, San Diego, CA, USA). The Shapiro–Wilk test was applied to assess data distribution. For non-normally distributed data, the Mann–Whitney U test was used to calculate the *p*-values, while Student’s *t*-test was employed for normally distributed data. An ANOVA with Bonferroni correction was used to compare more than two groups. Correlations were analysed in JASP (Department of Psychological Methods, University of Amsterdam, Amsterdam, The Netherlands), using Spearman’s test. Only the correlations with a *p* < 0.05 and an rho > 0.5 or an rho < −0.5 were deemed significant.

## 3. Results

### 3.1. The γδ T Cells in CLL Patients Had Higher LAG-3 and Lower CD16 Expression

First, the CLL patients were compared against healthy volunteers to assess the changes related to the disease. The CLL patients had significantly higher expression of CD56 and LAG-3 and significantly lower expression of CD16 on γδ T cells (Figure 2A–C). There was no difference in regard to CD69 and CD57 expression (Figure 2D,E).

### 3.2. CLL Patients Without Negative Prognostic Factors Had Higher Expression of CD69 and CD16

Next, we analysed the results in regard to the CLL patients subdivided into clinical subgroups. The patients were initially stratified based on the expression of two key prognostic markers, ZAP-70 and CD38, into positive and negative groups, based on the established 20% and 30% cut-off points, respectively [19]. The patients with an expression level above those thresholds, thus ZAP-70 or CD38 positive, are considered to be at high risk of disease progression. Interestingly, ZAP-70-negative patients exhibited higher expression of CD69 (Figure 3D), while the CD38-negative group showed increased expression of CD16 (Figure 3F). No significant differences were observed for the other markers (Figure 3A–C,E,G–J).

Afterwards, the patients were stratified according to Rai’s stages of disease. The only notable observation was higher expression of CD16 in stages III + IV (Figure 4A–E). Next, we divided the patients based on the treatment requirement during the observation period and survival during follow-up. Neither comparison showed any significant differences (Appendix A). Finally, we divided the patients based on *ATM* and/or *TP53* deletions, but no significant differences were observed between those groups (Appendix A).

### 3.3. CD16 Expression Inversely Correlated with Serum LDH

To better understand the clinical importance of the analysed markers, their expression was correlated against laboratory parameters (WBC #, LYMPH #, MONO #, NEUTR #, PLT #, LDH, IgA, IgG, IgM, β2-microglobulin, haemoglobin), patient age, time to treatment, and immunological parameters (the percentage of B cells [CD19+], CLL B cells [CD19+/CD5+]). Among these correlations, only CD16 expression on γδ T cells showed a significant inverse relationship with serum LDH levels (⍴ = −0.597, *p* < 0.001) (Figure 5). No other correlation was significant; the entire matrix is presented in Appendix A.

### 3.4. Patients with Unmutated IGVH Had Higher Expression of LAG-3 Ligands

Finally, to assess the importance of LAG-3 in regard to the pathogenesis of CLL, the expression of its ligands was evaluated. The Chronic Lymphocytic Leukemia (Broad, Nature 2015) RNAseq dataset was analysed using the cBioPortal in regard to the expression of class II HLA genes and the conventional LAG-3 ligands: *FGL1, LGALS3*. There were no data on the expression of *FGL1, HLA-DRB2, HLA-DRB3, and HLA-DRB4*. Interestingly, patients with unmutated *IGVH* had higher expression of *HLA-DQA2* and *HLA-DRB5* (Figure 6F,I).

## 4. Discussion

In the current study, we performed the first analysis of CD16, CD56, CD57, CD69, and LAG-3 expression on γδ T cells in CLL. Moreover, it is one of only a few studies focused on those receptors in regard to haematological malignancies. Most importantly, a significant and potentially clinically relevant dysregulation of CD16 and LAG-3 was observed.

The expression of CD16 on γδ T cells is essential from a therapeutic point of view. CD16+ γδ T cells are known to exert ADCC against multiple myeloma [20]. Moreover, rituximab, a therapeutic antibody widely utilised in the treatment of B cell malignancies, is known to activate human γδ T cells effectively [21]. In our study, CD16+ γδ T cells were decreased in the CLL patients; moreover, lower CD16 expression was noted in patients with negative prognostic markers (CD38+ patients). Additionally, the expression of CD16 correlated inversely with LDH. LDH is an indirect marker of CLL proliferation and advancement; elevated LDH suggests a more advanced and aggressive disease [22,23]. CD16 expression is regulated, at least partially, by the cytokine milieu; Chen and Freedman observed that it is upregulated in response to IL-2 and IL-15 [24]. On the other hand, cytotoxic T cells tend to downregulate CD16, while differentiating from naive towards effector cells [25]. Interestingly, while CD16 was decreased in high-risk patients (CD38+), we also observed an increase in Rai’s III-IV stage. While this seems contradictory, it is most probably related to the change in the Vδ1/Vδ2 ratio. We have previously reported that the amount of Vδ1 usually decreases with disease progression [15]. As Fisher et al. noted, with a higher density of CD16 expression on Vδ2 than Vδ1 cells, the decrease in Vδ1 may lead to the increase in CD16+ γδ T cells [26].

The overexpression of LAG-3 on γδ T cells was previously reported in regard to numerous cancers and during chronic inflammation, for e.g., melanoma or Plasmodium vivax infection [27,28,29,30,31]. Interestingly, only a minor upregulation was noted in melanoma patient blood [28,30]. This contrasts with a very high upregulation observed in the current study; depending on the patient, even above 90% γδ T cells expressed LAG-3. This partially corresponds to the results obtained for tumour tissue from melanoma patients [28]. Altogether, this suggests that such an increase in LAG-3 results from prolonged inflammation and constant stimulation with tumour cells. Unfortunately, this should also be viewed as a sign of exhaustion. Interestingly, Lin et al. suggest that short-term incubation of γδ T cells at 40 °C downregulates LAG-3 and PD-1, while upregulating IFN-γ [32]. The latter observation is interesting from an immunotherapeutic point of view, it suggests not only the possibility of increasing the fitness of in vitro generated γδ T cells, but also the potential reversibility of the exhaustion-like phenotype of γδ T cells observed in numerous cancers, including CLL. LAG-3 is usually highly expressed in leukemic B cells and, to a lesser extent, by T cells in CLL [33,34]. Still, the current study is the first to assess LAG-3 on γδ T cells in CLL and one of only a handful of such studies in general. The high expression of LAG-3 is linked to a significantly shorter time to the first treatment, suggesting its clinical importance [33]. Interestingly, the in vitro blockage of LAG-3 by a monoclonal antibody, relatlimab, in CLL patient-derived PBMCs improved NK cell cytotoxicity and the production of IL-2 by conventional T cells [35]. Finally, CLL cells usually exhibit high levels of the main LAG-3 ligand, namely galectin-3 [36]. Thus, the co-administration of the anti-LAG-3 antibody with in vitro expanded γδ T cells could be a new viable option for treating CLL.

Classically, CD69 is viewed as one of the early activation markers, but it is also an important regulator of cytokine production (including IFN-γ) and tissue retention [37]. Higher CD69 expression by γδ T cells is linked with better survivability in regard to hepatocellular carcinoma [38]. Interestingly, CD25-CD69+ Th cells are sometimes considered to have Treg-like potential. Their accumulation was observed in leukemic patients; moreover, their percentage was significantly higher in patients who had a relapse after haematopoietic stem cell transplantation [39]. While this observation has not been directly confirmed in human γδ T cells, it should be considered, especially since human γδ T cells may have a Treg-like phenotype [40]. CD69 is also a ligand of the inhibitory B7-H3 molecule; the inhibition of B7-H3 significantly improves the cytotoxic response of human γδ T cells [41]. Thus, CD69 should be examined cautiously; its high expression may suggest γδ T cell activation, but may also indicate their possible exhaustion or immunosuppressive potential.

Gabrielle Siegers et al. showed that CD56 expression on γδ T cells, especially the Vδ1 subset, correlates with their cytotoxic activity against CLL cells [42]. Similarly, γδ T cells obtained from G-CSF mobilised donors showed high expression of CD56 and significant cytotoxicity against neuroblastoma (line NB1691) and erythroleukemia (like K562) [43]. We observed a relatively high expression of CD56 on γδ T cells from CLL patients. This may indicate an initial response against CLL clones. However, the markedly elevated LAG-3 expression suggests potential exhaustion resulting from prolonged exposure to tumour antigens, indicating a dynamic transition from early activation to immune dysfunction. Nevertheless, this suggests that additional external aid, for e.g., the anti-LAG-3 antibody, could significantly increase the natural response against CLL cells.

CD57 expression on T cells is linked with cytotoxic potential and the anti-viral response [44]. Although the CD57+ CD8 cells are considered senescent and proliferate poorly, they retain high cytotoxic potential and are not necessarily exhausted [45,46]. Nevertheless, an accumulation of CD57+ Th lymphocytes was noted in various cancers, including Hodgkin’s lymphoma and CLL [47,48,49]. Although the percentage of CD57+ Th cells tends to increase with CLL progression, only some insignificant trends could be noted in our data. This suggests that in contrast to conventional Th lymphocytes, the expression of CD57 by γδ T cells does not necessarily yield negative implications.

The current study introduces an important novelty, most importantly, the exceptionally high expression of LAG-3 on γδ T cells. Nevertheless, it also has significant limitations. It is a cross-sectional study, performed in a single centre; some of the observed changes could be specific to the Polish population. More importantly, this study is observational in nature, and its clinical significance is based on correlative data. Thus, further explorations, with functional analyses and encompassing other populations, hopefully also non-Caucasians, seem necessary to fully comprehend the changes. Moreover, while we provide an extensive characterization of selected cytotoxicity-related markers on γδ T cells in CLL, we acknowledge that the analysis does not encompass other key activating receptors, such as DNAM-1 or NKG2D, or inhibitory molecules like PD-1, which have been implicated in cell exhaustion and immune surveillance. Future studies incorporating these markers would provide a more comprehensive view of γδ T cell dysfunction in CLL. Finally, while our findings suggest functional alterations in γδ T cells, this study is primarily observational, and functional assays assessing cytotoxicity, cytokine production, and exhaustion profiles were not performed.

## 5. Conclusions

CLL negatively affects γδ T cells. Most importantly, the prolonged inflammation and high tumour burden of CLL lowers CD16 expression, while also upregulating that of LAG-3. The latter seems to be especially clinically relevant; a blockage of LAG-3 may considerably increase the cytotoxic response of γδ T cells. LAG-3 holds significant interest to warrant further investigation of its role in γδ T cell effector functions in CLL, which could eventually lead to evaluating its potential for therapeutic applications. The relatively high expression of CD56 suggests that circulating γδ T cells have significant cytotoxic potential in CLL, but that they require external stimulation.

## Figures and Tables

**Figure 1 cells-14-00451-f001:**
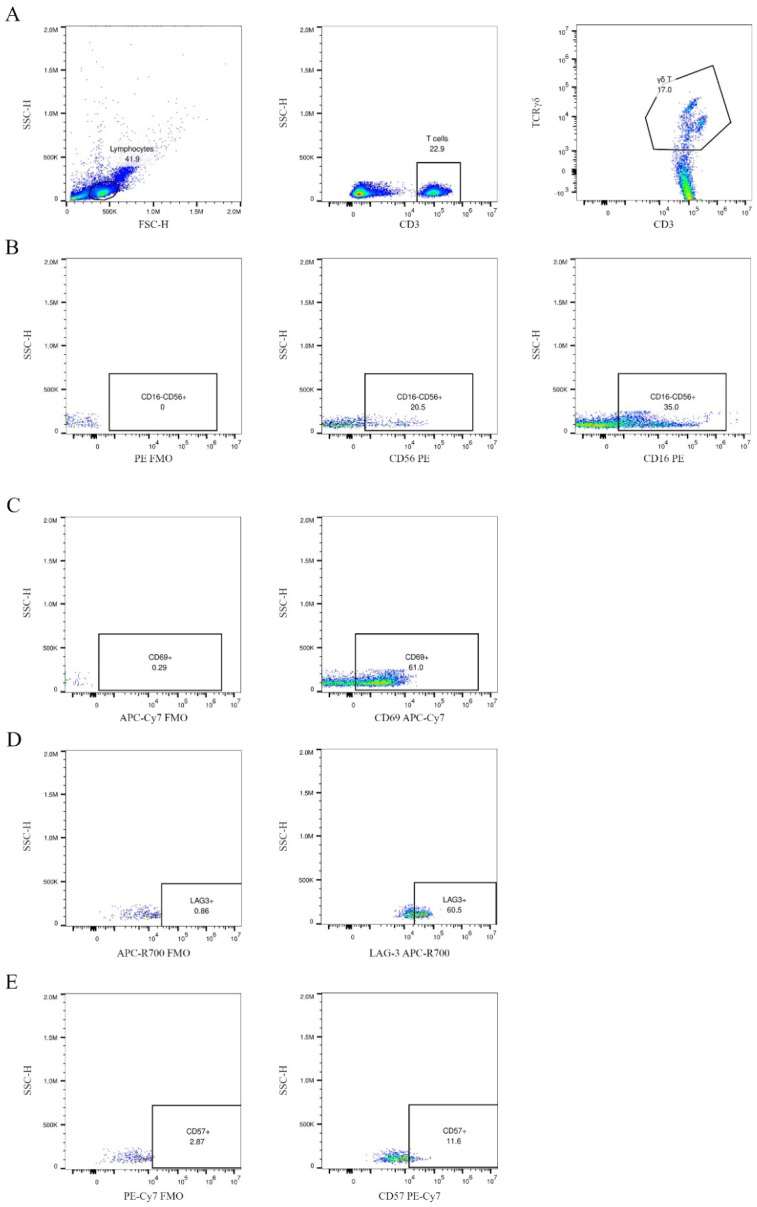
Gating strategy for γδ T cells and selected surface markers. Specifically, γδ T cells were identified within the total T cell population (CD3+, stained with APC anti-CD3) using FITC anti-TCRγδ (**A**). The expression of CD16, CD56, CD57, CD69, and LAG-3 on γδ T cells was evaluated using two separate staining panels. Panel 1 included CD56 PE, CD57 PE-Cy7, and LAG-3 APC-R700; panel 2 included CD16 PE and CD69 APC-Cy7. Fluorescence-minus-one (FMO) controls were used for each channel to ensure accurate gating. In (**B**–**E**), the first dot plot in each row represents the FMO control, followed by the specific staining for each marker. The dot plots present a CLL patient at stage I, according to Rai’s criteria, namely CD38 and ZAP-70 negative.

**Figure 2 cells-14-00451-f002:**
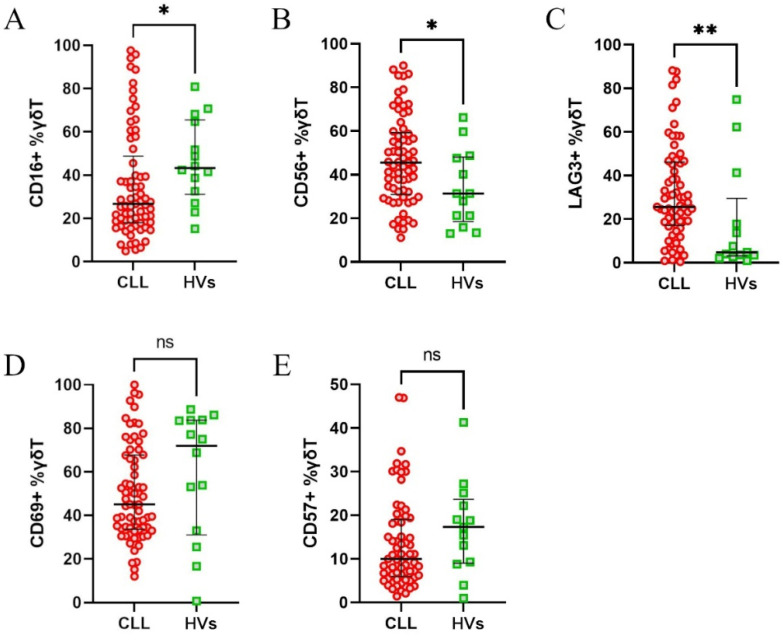
Differential expression of surface markers on γδ T cells from healthy volunteers (HVs) and CLL patients**.** The expression of CD16 (**A**), CD56 (**B**), LAG-3 (**C**), CD69 (**D**), and CD57 (**E**) on γδ T cells was assessed. The data are presented as percentages of γδ T cells expressing the respective markers. The CLL patients exhibited significantly higher expression of CD56 and LAG-3, but lower expression of CD16, compared to the HVs (**A**–**C**). No significant differences were observed for CD69 and CD57 (**D**,**E**). Statistical significance was calculated using the Mann–Whitney U test, with * *p* < 0.05, ** *p* < 0.01, and “ns” indicating not significant.

**Figure 3 cells-14-00451-f003:**
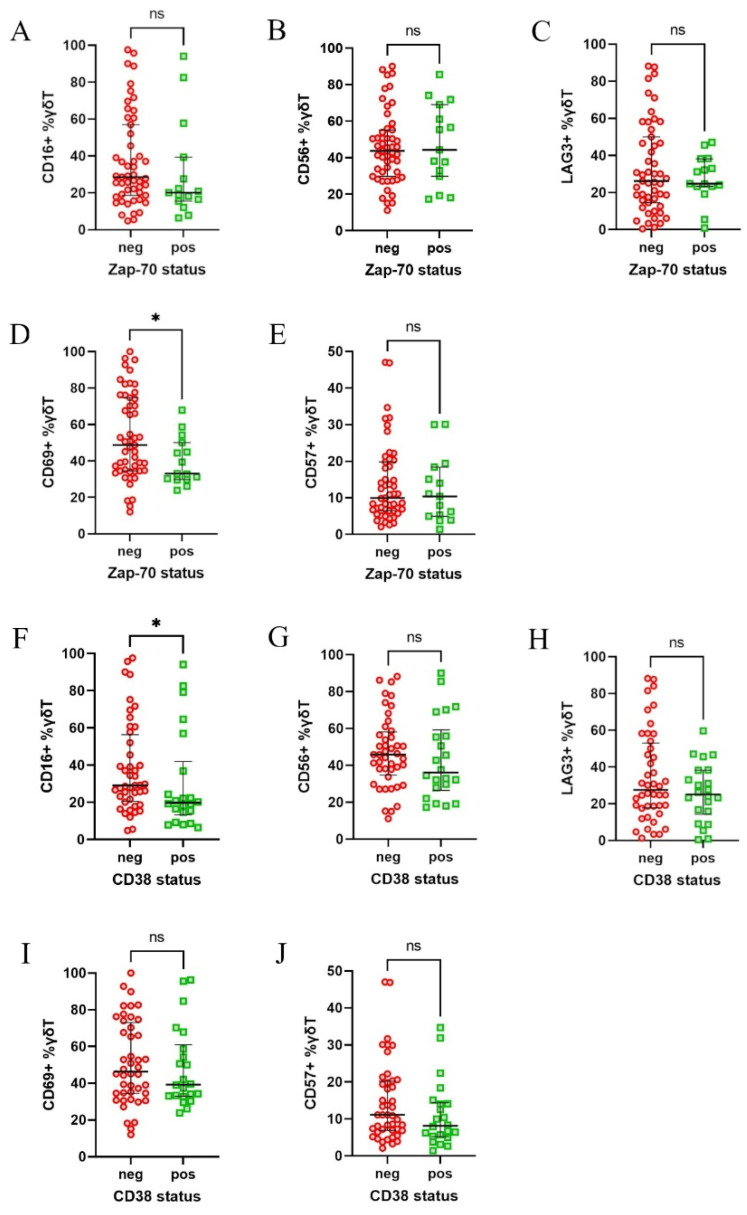
Differential expression of CD16, CD56, CD57, CD69, and LAG-3 on γδ T cells in CLL patients, divided according to their ZAP-70 and CD38 status. Patients were divided into ZAP-70-positive (≥20% ZAP-70+ neoplastic cells) and ZAP-70-negative groups, as well as CD38-positive (≥30% CD38+ cells) and CD38-negative groups. CD69 expression was significantly higher in ZAP-70-negative patients (**D**), while CD16 expression was significantly increased in the CD38-negative group (**F**). No significant differences were observed for the other markers in regard to either ZAP-70 (**A**–**C**,**E**) or CD38 (**G**–**J**) stratifications. Statistical significance was assessed using the Mann–Whitney U test, with * *p* < 0.05 and “ns” indicating not significant (*p* > 0.05). Abbreviations: neg, negative; pos, positive.

**Figure 4 cells-14-00451-f004:**
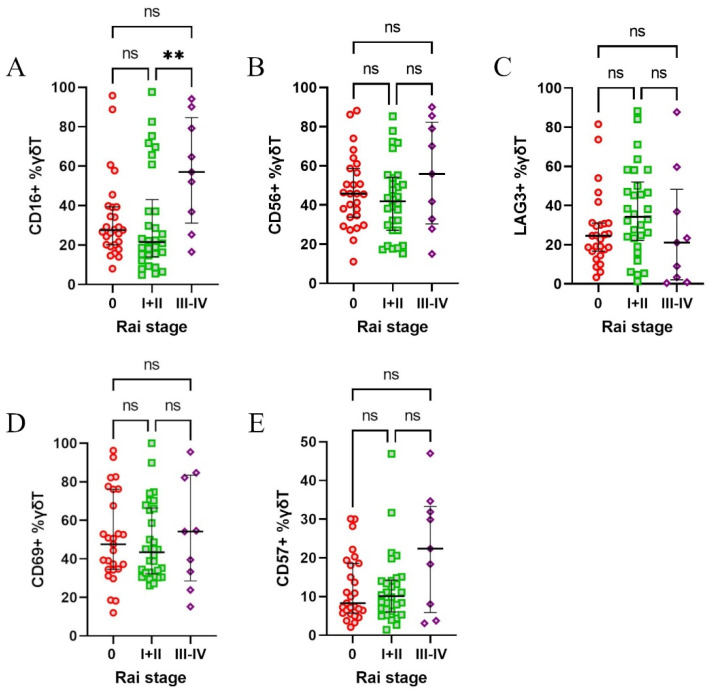
Expression of CD16, CD56, CD57, CD69, and LAG-3 on γδ T cells in CLL patients stratified according to Rai’s stages of disease**.** Patients were grouped into low-risk (Rai stage 0), intermediate-risk (Rai stages I and II), and high-risk (Rai stages III and IV) categories, based on the disease progression risk. CD16 expression was significantly higher in high-risk patients (stages III and IV) compared to intermediate-risk patients (stage I and II) (**A**). No significant differences were observed for CD56 (**B**), LAG-3 (**C**), CD69 (**D**), or CD57 (**E**) across Rai’s stages. Statistical significance was assessed using the Kruskal–Wallis test, with, ** *p* < 0.01, and “ns” indicating not significant (*p* > 0.05).

**Figure 5 cells-14-00451-f005:**
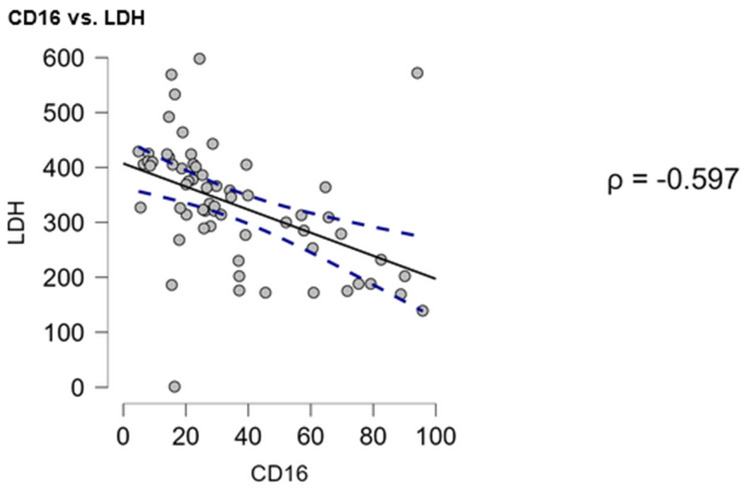
Correlation between CD16 expression on γδ T cells and serum LDH levels in CLL patients. Spearman’s rank correlation analysis revealed a significant inverse relationship between CD16 expression on γδ T cells and serum LDH levels (⍴ = −0.597, *p* < 0.001). The solid line represents the best-fit regression line, and the dashed lines indicate the 95% confidence interval.

**Figure 6 cells-14-00451-f006:**
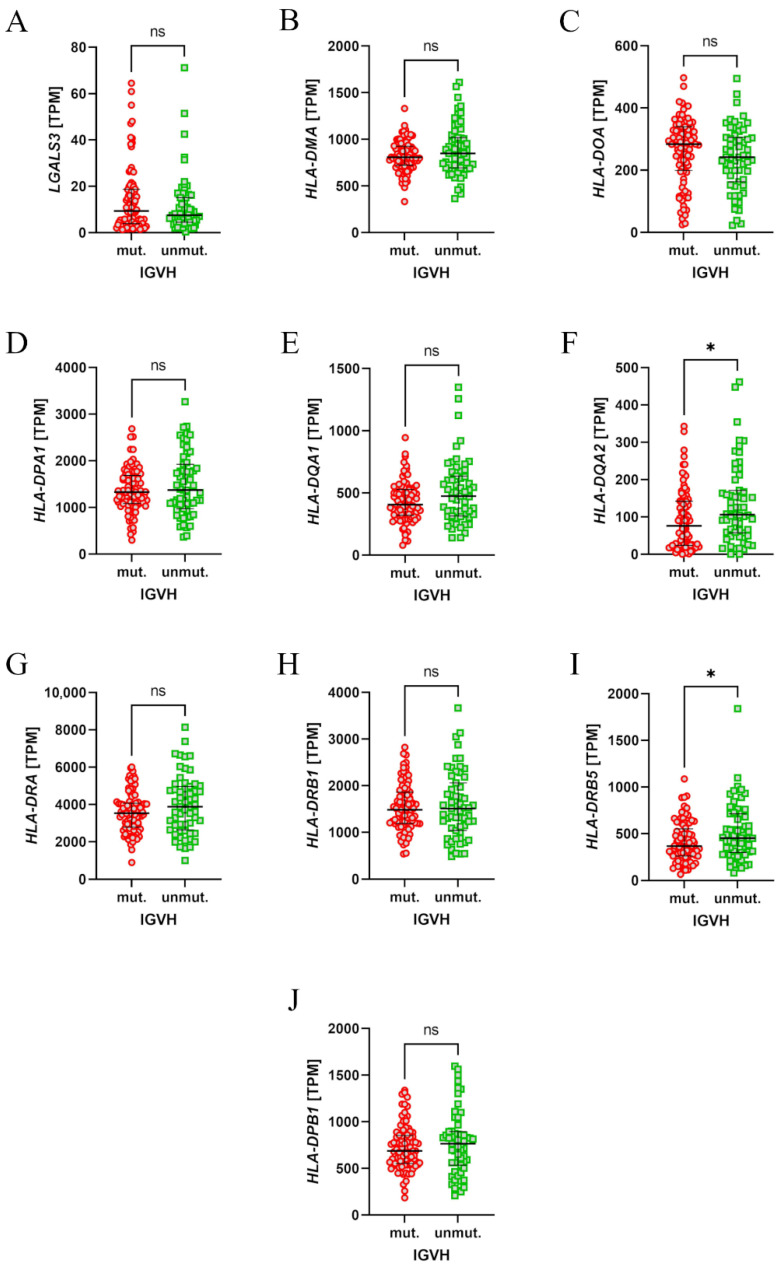
The expression levels of the LAG-3 ligand mRNA in neoplastic B cells from patients with chronic lymphocytic leukaemia (CLL). Mutated (mut.) and unmutated (unmut.) *IGVH* groups are compared for each ligand; expression is measured in transcripts per million (TPM). Significant differences are indicated by * (*p* < 0.05), while “ns” denotes no significant differences (*p* > 0.05).

## Data Availability

The data generated and analysed during this study are available from the corresponding author upon reasonable request.

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
