# Peer review of "Characterisation of Cytotoxicity-Related Receptors on γδ T Cells in Chronic Lymphocytic Leukaemia"

_cells, 2025, doi:10.3390/cells14060451_

Round 1
Reviewer 1 Report
Comments and Suggestions for Authors
As a molecular immunologist, from the abstract, I felt the MS would shed some light on identifying novel markers. After carefully reading, I realized the authors made an effort to characterize the gd T cell population in CLL patients.
Authors measure important markers such as CD16, CD56, CD57, CD69 and LAG-3 on total peripheral gd T cells. The data presented is novel with regard to the CLL patient dataset. It would have been very encouraging to see the data of two distinct and important peripheral gd T cell subsets, Vg9Vd2 T and Vd1+ T cells.
Additionally, among the CD16, CD56 and LAG-3, differences in LAG-3 expression seem promising to be elevated as a marker for further study in diagnostics to diagnose a CLL patient. Despite that, I find that the numbers of markers evaluated, and details of the gd T cell subsets involved are not sufficient for publication,
I would recommend the preparation of the manuscript with further markers like NK-like receptors, including NKG2D, DNAM-1 and other important effector function proteins like granzyme and perforin to be included. It is appropriate to dissect the expression of those markers in different above-mentioned gd T cell subsets. Also, possibly patient-derived gd T cell subsets, Vg9Vd2 T and Vd1+ T cells coculture with CLL cell lines would reveal the functional efficacy as well.
Figure 1 gating strategy – is it a representative of a healthy volunteer (HV) or CLL patient? Is there any difference in the percentage of gd T-cells between HV and CLL? The Authors shall mention whether the total gd T cell percentage was comparable between HV and CLL.
- Line 42, insert reviews to gd T cell diversity and antigens recognised. Vantourout et al., 2013 Six of the best: Unique contribution of gd T cells to Immunology; Castro et al., 2020 Diversity in recognition and function of gd T cells.
- Alongside introducing CD16, CD56, CD57 and CD69, authors should also introduce LAG-3, its function and significance.
- Figure 2, though CD16 and CD56 show significant differences between CLL and HV, I find LAG-3 differences stronger and may be evaluated in the diagnosis.
- In Line 121, the authors mentioned ZAP70 and CD38-based segregation of population and here, authors must mention the reason and their clinical relevance.
- Was there any correlation of CD16 expression in Fig 5 patients categorised based on Rai recommendation to CD16 expression shown in Fig 4, can this be further categorized like ZAP70 and CD38 pos and neg patients of Rai stages?
- In Figure 6, Fig 6C and Fig 6D, are identical, is it a mistake in labeling of graphs and duplications?
- Lines 216-219 describing Th cells seem irrelevant to this context
- 226 – 227, leaves out the meaning that CD56+ gd T cells are the ones that show exhaustion phenotype, however, in this study overall gd T cell population is measured for markers, not a distinct population evaluated for phenotype. So, it shall be rewritten something like, “γδ T cells in CLL patients exhibited higher CD56 expression, suggesting an initial response against CLL clones. However, the significantly elevated LAG-3 expression indicates potential exhaustion due to prolonged exposure to tumour antigens, highlighting a dynamic shift from early activation to exhaustion.”
Quality of the writing is fine
Reviewer 2 Report
Comments and Suggestions for Authors
Overall: This study investigated the expression of cytotoxicity-related receptors, CD16, CD56, CD57, CD69 and checkpoint (LAG-3) on γδ T cells in CLL patients. Very detailed flow gating strategy, providing great guidelines for studies on γδ T cell in CLL, with good sample size. Relatively fewer studies focusing on γδ T in CLL and this study enriched understanding of γδ T cells.
Comments:
Figure 1 A: Is this panel representing for CLL patients or healthy individuals? The percentage of of lymphocytes seem oddly low if this is from CLL patients. The total lymphocytes in PBMC from CLL patients are typically high.
Result 3.2: Have authors checked the correlation between CD40d subgroups and these receptors? CD38, ZAP-70 and CD49d are all important prognostic markers and CD49d is considered as the strongest predictor of poor outcome with current studies.
How about the correlation of FISH status to these markers?
Reviewer 3 Report
Comments and Suggestions for Authors
This manuscript investigates the expression of cytotoxicity-related receptors (CD16, CD56, CD57, CD69) and the immune checkpoint molecule LAG-3 on γδ T cells in CLL patients. Given the important role of γδ T cells in tumor immunity and their potential for engineered T-cell therapy, this study provides valuable insights. The manuscript is well-written, with a detailed analysis integrating flow cytometry data and clinical parameters. However, the following concerns should be addressed to strengthen the study:
-
The authors report low CD16 expression on γδ T cells in CLL patients compared to healthy donors and in CD38+ CLL patients compared to CD38− patients, yet they also find higher CD16 expression in patients with high-risk Rai stage (Stages III-IV) compared to low-risk Rai stage. How do the authors reconcile these findings?
-
The rationale for investigating only LAG-3 as an immune checkpoint marker should be explained. Were other checkpoint molecules considered, and if so, why were they not included in the analysis?
-
In lines 120-122, the authors introduce clinical cut-off values for ZAP-70 and CD38 in CLL patients. To aid reader comprehension, it would be helpful to explicitly state that patients above these cut-offs are classified as high-risk.
-
In Figure 5, the x-axis label is missing and should be added for clarity.
Round 2
Reviewer 1 Report
Comments and Suggestions for Authors
I appreciate that the authors have addressed the minor comments in the manuscript. While the overall message of elevated LAG-3 expression in CLL patients is novel, the analysis could be strengthened by expanding the range of γδ T cell markers. Given the observed increase in LAG-3 expression in CLL patients compared to healthy volunteers, it would be valuable to explore its impact on γδ T cell effector functions in future studies.
I recommend the manuscript for publication as a short communication. However, I strongly suggest revising the following statement in the conclusions, as it is too speculative: “LAG-3 should therefore be considered as a potential therapeutic target in CLL.” The current data highlight LAG-3 expression, which, due to its inhibitory nature, may play a role in negatively regulating T-cell responses. I suggest rephrasing the statement to:
"LAG-3 holds significant interest for further investigation of its role in gd T cell effector functions in CLL, which could eventually lead to evaluating its potential for therapeutic applications."
Author Response
Thank you for your insightful suggestions. As per your recommendation, we have revised the conclusion to make it less speculative. The updated statement now reads:
"LAG-3 holds significant interest for further investigation of its role in γδ T cell effector functions in CLL, which could eventually lead to evaluating its potential for therapeutic applications."